

# DMRG investigation of constrained models: from quantum dimer and quantum loop ladders to hard-boson and Fibonacci anyon chains

**Natalia Chepiga**[1][*][†] **and Frédéric Mila**[2]

**1** Department of Physics and Astronomy, University of California, Irvine, CA, USA
**2** Institute of Physics, École Polytechnique Fédérale de Lausanne (EPFL),
CH-1015 Lausanne, Switzerland

[*] natalia.chepiga@alumni.epfl.ch
[†] Current address: Institute for Theoretical Physics , University of Amsterdam,
Science Park 904, 1098 XH Amsterdam, The Netherlands

## Abstract

**Motivated by the presence of Ising transitions that take place entirely in the singlet sector of frustrated spin-1/2 ladders and spin-1 chains, we study two types of effective dimer models on ladders, a quantum dimer model and a quantum loop model. Building on the constraints imposed on the dimers, we develop a Density Matrix Renormalization Group algorithm that takes full advantage of the relatively small Hilbert space that only grows as Fibonacci number. We further show that both models can be mapped rigorously onto a hard-boson model first studied by Fendley, Sengupta and Sachdev [Phys. Rev. B 69, 075106 (2004)], and combining early results with recent results obtained with the present algorithm on this hard-boson model, we discuss the full phase diagram of these quantum dimer and quantum loop models, with special emphasis on the phase transitions. In particular, using conformal field theory, we fully characterize the Ising transition and the tricritical Ising end point, with a complete analysis of the boundary-field correspondence for the tricritical Ising point including partially polarized edges. Finally, we show that the Fibonacci anyon chain is exactly equivalent to special critical points of these models.**


# 1   Introduction

Frustrated quantum magnetism is one of the most challenging and actively developing fields of condensed matter physics. Classical examples of long-range order in magnets are ferromagnetic and Néel order, but the theoretical and experimental investigation of many models and compounds has revealed a large variety of long-range orders due to quantum effects. Traditionally the theoretical investigation of quantum magnetism relies on simple models, typically Ising or Heisenberg models. In 1988, Rokhsar and Kivelson proposed a model of interacting hard-core dimers [1] to describe the resonating-valence bond (RVB) phase of spin-1/2 models [2–4] in the context of the high-temperature cuprate superconductors. In this approach, hard-core dimers are used to represent SU(2) invariant spin singlets. This formulation of the model of hard dimers goes under the name of quantum dimer model (QDM).

In a pictorial representation, the quantum dimer model is defined by the following Hamiltonian:

$$H_{\text{QDM}} = \sum_{\text{Plaquettes}} \left[ -J \left( \left| \updownarrow \updownarrow \right\rangle \left\langle \leftrightarrows \right| + \text{h.c.} \right) + v_{\text{rung}} \left| \updownarrow \updownarrow \right\rangle \left\langle \updownarrow \updownarrow \right| + v_{\text{leg}} \left| \leftrightarrows \right\rangle \left\langle \leftrightarrows \right| \right], \qquad (1)$$

where $J > 0$ is the coupling constant of a kinetic term that flips pairs of dimers on a plaquette, and $v_{\text{rung}}$ ($v_{\text{leg}}$) is the coupling constant of a potential term that counts the total number of flippable plaquettes with dimers on vertical (horizontal) bonds. In the isotropic two-dimensional QDM, $v_{\text{rung}} = v_{\text{leg}}$, and the potential term just counts the number of flippable plaquettes with equal weight.

In general, the bulk ground-state of an antiferromagnet is given by a singlet state. To define a good variational subspace, it is often useful to complement this global constraint by a local one according to which spins are paired and form dimer singlets with one of their neighbors, leading to a small energy when all spins are paired. Searching for all possible ways of pairing

spins corresponds to the problem of finding all possible dimer coverings on a given lattice. The RVB state is defined as a superposition of all possible dimer coverings.

Our motivation to study the QDM on two-leg ladders comes directly from the frustrated spin-1/2 ladders with $J_1 - J_2$ interactions on legs. This model undergoes transition between the rung-singlet phase and the columnar dimer phase that has been reported to be continuous in the Ising universality class [5]. This phase transition occurs entirely in the singlet sector, and the singlet-triplet gap remains open at the transition. This suggests that the same critical line could also appear if one focuses on a QDM with a Hilbert space of nearest-neighbor singlets. In the same spirit, for the $J_1 - J_2$ spin-1 chain with three-site [6] or biquadratic [7] interaction, it has been shown that the transition between the next-nearest-neighbor Haldane phase and the dimerized phase is also Ising and occurs entirely in the singlet sector, with a singlet-triplet gap that remains open as in the previous case. In the Affleck-Kennedy-Lieb-Tasaki (AKLT) valence-bond picture [8], two spin-1/2 dimer singlets emanate from every site, and the corresponding constrained model is a quantum loop model (QLM) [9] in which a spin-1 singlet dimer is associated with a trivial loop of length 2 (see below). The possibility to access such transitions in the context of simpler models that only describe the singlet sector is interesting in itself as a confirmation of the non-magnetic nature of the transition. It also opens the way to the investigation of more complicated models in which a similar physics is expected to take place, for instance frustrated spin-S chains with S>1.

An important difference between spin models and the QDM or QLM comes from the definition of the elementary degrees of freedom. While spin degrees of freedom are located at the nodes of the lattice, the dimer degrees of freedom are associated with the bonds between the sites. Besides, there is a local constraint that specifies the number of dimers emanating from a given site. The associated Hilbert spaces are thus given by the complete set of fully packed dimer or loop configurations, with a dimension that grows much more slowly with the number of sites $N$ than $2^N$ resp. $3^N$ for spin-1/2 or spin-1 models.

Along with the QDM and QLM, there are other constrained models that have attracted the attention of condensed matter physicists over the past decades. Indeed, apart from their general mathematical interest, constrained models are convenient toy models for many physical problems. By reducing the Hilbert space around some particular constrained basis one can focus on the essential properties of the low-energy sector, discarding less important higher-energy effects. As we shall see, two of them are closely related to the QDM and QLM: a hard-boson model introduced by Fendley et al [10], in which bosons are constrained to sit neither on the same site nor on neighboring sites, and a model of interacting anyons also known as the Fibonacci chain [11], in which the Hilbert space is constructed according to fusion rules that lead to local exclusions. In fact, we will show that all these models can be mapped onto each other.

The rest of the paper is organized as follows. In Section 2, we give a detailed presentation of all the models mentioned above (QDM, QLM, hard boson and anyons) and of the relation between them, starting with the hard-boson model of Fendley et al [10] that will serve as a reference throughout. In Section 3 we discuss the implementation of the quantum dimer constraints in the Density Matrix Renormalization Group (DMRG) algorithm. Section 4 is devoted to the phase diagram of the QDM on a simple two leg ladder, and of the equivalent hard-boson model. It includes an overview of the transition between the rung-dimer and the period three phases, and a numerical investigation of the boundary-field correspondence at the Ising tricritical point. In section 5, we discuss implications of these results for the phase diagram of the QLM on a zig-zag ladder. Section 6 summarizes our main results.

## 2 Models and mappings

### 2.1 Hard-boson model

In Ref. [10], Fendley et al. have studied a model of hard bosons $d_i^\dagger, d_i$, i.e.. bosons that satisfy not only the constraint of hard-core bosons, $n_j(1-n_j)=0$, but also the additional constraint that they cannot sit on neighboring sites, i.e. the constraint $n_j n_{j+1}=0$. Their model is defined by the following Hamiltonian:

$$H_{\text{HB}} = \sum_j \left[ -w(d_j^\dagger + d_j) + U n_j + V n_{j-1} n_{j+1} \right]. \tag{2}$$

They have shown that this model has three main phases: a disordered phase that does not break the translation symmetry, a charge density wave of period 2, and a charge density wave of period 3. The nature of the transition out of the period 3 phase is a subtle problem that has been recently revisited by Samajdar et al [12], and by the present authors [13] using the algorithm described below. A full summary of the current understanding of the phase diagram of this model will be given in Section 4.

### 2.2 Quantum Dimer Model

When defined on a two-leg ladder, the minimal version of the QDM given by Eq. 1 also has three types of simple ground states. When $v_{\text{rung}}/J \to -\infty$ and $v_{\text{leg}}/J \to 0$, the energy is minimized by the state with a maximal number of flippable plaquettes with vertical dimers. It corresponds to the rung dimer state sketched in Fig.1(a). By contrast, in the limit $v_{\text{leg}}/J \to -\infty$ and $v_{\text{rung}}/J \to 0$ the energy is minimized by the state with a maximal number of flippable plaquettes with horizontal dimers. Due to the quantum dimer constraint no more than one of two consecutive plaquettes can carry leg dimers. It corresponds to a columnar phase in which leg dimers facing each other occupy every other plaquette. One of these states is sketched in Fig.1(b). When $v_{\text{rung}}/J = v_{\text{leg}}/J \to +\infty$ the system will minimize the number of flippable plaquettes. On an infinite system, or on clusters with periodic boundary conditions, this is achieved by the two staggered states in which all plaquettes are non-flippable (see Fig.1(c)). These two states are not connected to the rest of the Hilbert space by the Hamiltonian of Eq.1, and more generally by any local Hamiltonian. They constitute two separate sectors of the Hilbert space, and they have zero energy for all parameters. If we exclude these two states from the Hilbert space, the number of flippable plaquettes is minimized by the state that has a sequence of rung dimer and flippable plaquettes with two dimers on the legs sketched in Fig.1(d), so that every third plaquette is flippable with dimers on legs. Note that, on clusters with open boundary conditions and vertical edges, there is no need to exclude the staggered states from the Hilbert space since they are incompatible with the boundary conditions. In the rest of the paper, we will concentrate on the QDM defined on a Hilbert space from which non-flippable states (if any) are excluded.

(a)    (b)    (c)    (d)

Figure 1: Sketches of (a) the rung dimer state, (b) one of the two columnar states, (c) one of the two staggered states, and (d) one of the three period-three states. For (b), (c) and (d), the other states can be obtained by translation.

This model can be mapped onto a hard boson model living on the plaquette in the following way. To associate a hard boson configuration to a dimer configuration, put a boson on each

plaquette with horizontal legs, and no boson on any other type of plaquette (see Fig. 2). Since two plaquettes with horizontal legs cannot be nearest neighbors, the resulting boson configuration satisfies the constraints of hard bosons. Conversely, any configuration of hard bosons can be associated to a unique dimer configuration according to the following rules: (i) Put a rung dimer between any pair of adjacent empty sites; (ii) Put two leg dimers on the plaquette of any occupied site. These rules follow easily from the dimer constraint. So there is a one to one correspondence between the Hilbert spaces of the two models.

The three operators entering the QDM model of Eq. 1 can be easily written in term of hard-boson operators:

$$|{\rlap{\rule[0.1em]{0.01em}{0.7em}}\phantom{l}}\;{\rlap{\rule[0.1em]{0.01em}{0.7em}}\phantom{l}}\rangle\langle{\rule{0.7em}{0.01em}}| + |{\rule{0.7em}{0.01em}}\rangle\langle{\rlap{\rule[0.1em]{0.01em}{0.7em}}\phantom{l}}\;{\rlap{\rule[0.1em]{0.01em}{0.7em}}\phantom{l}}| = d_i^\dagger + d_i \tag{3}$$

because this term creates or destroys a pairs of neighboring dimers on the legs,

$$|{\rule{0.7em}{0.01em}}\rangle\langle{\rule{0.7em}{0.01em}}| = n_i \tag{4}$$

because this term only counts the plaquettes with leg dimers, and

$$|{\rlap{\rule[0.1em]{0.01em}{0.7em}}\phantom{l}}\;{\rlap{\rule[0.1em]{0.01em}{0.7em}}\phantom{l}}\rangle\langle{\rlap{\rule[0.1em]{0.01em}{0.7em}}\phantom{l}}\;{\rlap{\rule[0.1em]{0.01em}{0.7em}}\phantom{l}}| = (1 - n_{i-1})(1 - n_i)(1 - n_{i+1}) \tag{5}$$

because this term counts the sets of three consecutive plaquettes with no leg dimers.

Using the constraint $n_i n_{i+1} = 0$, the last term can be rewritten

$$|{\rlap{\rule[0.1em]{0.01em}{0.7em}}\phantom{l}}\;{\rlap{\rule[0.1em]{0.01em}{0.7em}}\phantom{l}}\rangle\langle{\rlap{\rule[0.1em]{0.01em}{0.7em}}\phantom{l}}\;{\rlap{\rule[0.1em]{0.01em}{0.7em}}\phantom{l}}| = 1 - n_{i-1} - n_i - n_{i+1} + n_{i-1}n_{i+1}. \tag{6}$$

So, up to a constant, the QDM can be written in terms of hard bosons as:

$$H_{\text{QDM}}^{\text{HB}} = \sum_j \left[ -J(d_j^\dagger + d_j) + (v_{\text{leg}} - 3v_{\text{rung}})\, n_j + v_{\text{rung}}\, n_j n_{j+2} \right]. \tag{7}$$

| Quantum dimer model | Hard bosons | Quantum loop model | Ising | Fibonacci anyons |
|---|---|---|---|---|
| (dimer config) | 1 | (loop configs) | ↑ | $I$ |
| (dimer configs) | 0 | (loop configs) | ↓ | $\tau$ |

Figure 2: Mapping between QDM, hard-boson, quantum loop, and Ising models on two-leg ladder. At the tricritical Ising point these models have an exact mapping to the Fibonacci anyon model.

It is also straightforward to rewrite the Hamiltonian of Eq.2 in terms of quantum dimer operators:

$$H_{\text{HB}}^{\text{QDM}} = \sum_{\text{Plaquettes}} \left[ -\omega\left(|{\rlap{\rule[0.1em]{0.01em}{0.7em}}\phantom{l}}\;{\rlap{\rule[0.1em]{0.01em}{0.7em}}\phantom{l}}\rangle\langle{\rule{0.7em}{0.01em}}| + \text{h.c.}\right) + U|{\rule{0.7em}{0.01em}}\rangle\langle{\rule{0.7em}{0.01em}}| + V|{\rule{0.7em}{0.01em}}\;{\rule{0.7em}{0.01em}}\rangle\langle{\rule{0.7em}{0.01em}}\;{\rule{0.7em}{0.01em}}| \right], \tag{8}$$

where the last term can be rewritten using Eq.4 and 6:

$$|{\rule{0.7em}{0.01em}}\;{\rule{0.7em}{0.01em}}\rangle\langle{\rule{0.7em}{0.01em}}\;{\rule{0.7em}{0.01em}}| = |{\rlap{\rule[0.1em]{0.01em}{0.7em}}\phantom{l}}\;{\rlap{\rule[0.1em]{0.01em}{0.7em}}\phantom{l}}\rangle\langle{\rlap{\rule[0.1em]{0.01em}{0.7em}}\phantom{l}}\;{\rlap{\rule[0.1em]{0.01em}{0.7em}}\phantom{l}}| + 3|{\rule{0.7em}{0.01em}}\rangle\langle{\rule{0.7em}{0.01em}}|. \tag{9}$$

Expressed in terms of single-plaquette variables, the Hamiltonian of Eq. 2 reads:

$$H_{\text{HB}} = \sum_{\text{Plaquettes}} \left[ -\omega \left( |{\updownarrow}{\updownarrow}\rangle \langle \cdots| + \text{h.c.} \right) + V |{\updownarrow}{\updownarrow}\rangle \langle {\updownarrow}{\updownarrow}| + (U + 3V) |{\cdots}\rangle \langle {\cdots}| \right]. \tag{10}$$

This Hamiltonian obviously reduces to quantum dimer Hamiltonian of Eq.1 with coupling constants:

$$J = \omega; \qquad v_{\text{rung}} = V; \qquad v_{\text{leg}} = U + 3V. \tag{11}$$

Without loss of generality we will set $w = 1$ throughout the paper. Note that the model reduces to the isotropic quantum dimer model when the last two coupling constants are equal, i.e. along the line $V + U/2 = 0$.

## 2.3   Quantum loop model

QDMs are natural effective models for spin-1/2 systems in the valence-bond basis. The valence bond basis is also very useful to discuss the properties of spin-1 chains, but in that case, a spin 1 is first split into two spins 1/2, and two valence-bond singlets emanate from each site. The resulting model in terms of dimers is then better described as a quantum loop model, with a Hilbert space consisting of all possible coverings of the lattice with non-overlapping loops in which each site belongs to one and only one loop. Since it is possible for two spins 1 to build a singlet, trivial loops consisting of a double dimer connecting the same pair of sites have to be included.

Such models can be useful in describing transitions in spin-1 models that take place entirely in the singlet sector, for instance the transitions between the next-nearest neighbor (NNN) Haldane phase and dimerized or trimerized phases, as reported in various frustrated spin-1 chains [6,7,14]. These phases are sketched in Fig.3(a-c). Since these frustrated chains include a next-nearest neighbor coupling, they are better seen as zig-zag chains, or alternatively as frustrated ladders, with two chains and a zig-zag coupling between them.

So, in the following, we will concentrate on the effective model defined by the following Hamiltonian:

$$H_{\text{QLM}} = -\sum_j \left[ (r_j^\dagger r_{j+2}^\dagger l_j l_{j+1} + \text{h.c.}) + \delta r_j^\dagger r_j^\dagger r_j r_j + \frac{\theta}{2} (r_j r_j^\dagger r_j^\dagger r_j + \text{h.c.}) \right], \tag{12}$$

where $r^\dagger$ and $l^\dagger$ create dimers on rungs and legs respectively. In a pictorial representation Eq.12 takes the following form:

$$H_{\text{QLM}} = -J \sum_{\text{Plaquettes}} \left( |{\slash}{\slash}\rangle \langle \cdots| + \text{h.c.} \right) - \sum_{\text{Rungs}} \left[ \delta |{\parallel}\rangle\langle{\parallel}| + \theta |{\slash}\rangle\langle{\slash}| \right], \tag{13}$$

where the first sum runs over all plaquettes with either orientation and with single or with double dimers on the rungs, while the second sum runs over all rungs with either orientation. The Hamiltonian of Eq.13 acts on a constrained Hilbert space in which each lattice node is connected by two and only two dimers with its nearest neighbors. Besides, for simplicity, we exclude the states with double dimers on the legs since they are not necessary to describe the transition between the next-nearest neighbor Haldane phase and the dimerized or trimerized phases [6,7,14]. Besides, we exclude the Haldane state where all nearest-neigbor rungs are occupied by a single dimer, as well as all the states derived from it by flipping plaquettes, for essentially the same reason that led us to exclude the staggered states in the QDM: they constitute a well separated sector, and they only exist on an infinite system or with periodic boundary conditions. With open boundary conditions, these configurations have to be excluded since

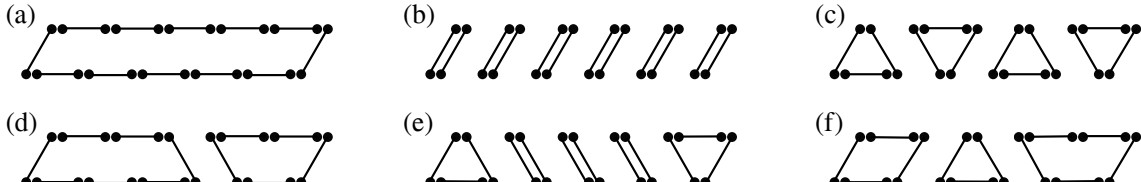

Figure 3: Sketches of quantum dimer ground state in (a) the NNN-Haldane; (b) the dimerized and (c) the trimerized phases. (d-f) examples of the other states that satisfy the quantum loop constraint

the edge sites would have only one dimer. In Fig. 3(d-f), we provide a few examples of states that satisfy the quantum loop constraint.

The first term of the Hamiltonian of Eq.12 is the straightforward generalization of the traditional kinetic term of the QDM to the Hilbert space with two local degrees of freedom. It flips pairs of dimers facing each other from rungs to legs and vice versa. The NNN-Haldane state sketched in Fig.3(a) is the ground-state of the Hamiltonian of Eq. 12 at $\delta = \theta = 0$, when only this term is present. The second term counts the number of rungs occupied by two dimers. In the limit $\delta \to \infty$, $\theta \to 0$, the ground state is the dimerized state with every other rung occupied by a double dimer, as sketched in Fig.3(b). The ground-state is two-fold degenerate and the translation symmetry is spontaneously broken. In the limit $\theta \to \infty$ the ground-state corresponds to the configuration with the maximal number of rungs occupied by a single dimer. Since the Haldane phase is excluded, this is achieved by the trimerized state, in which 2/3 of the nearest-neighbor bonds are occupied by a single dimer.

The QLM can be mapped on a model of hard bosons on the diagonal (zigzag) bonds of the ladder as follows. First of all, associate to any diagonal bond the two leg bonds with which it could build a triangle. Then, for any loop configuration, a diagonal bond is occupied by a boson ($n_i = 1$) if there is no dimer on that bond and on the associated leg bonds, and it is empty otherwise ($n_i = 0$). This is illustrated in Fig. 2. The constraint that two dimers have to start from every site implies that a site occupied by a boson will never have another occupied site as a neighbor. This construction thus associates to any loop configuration a configuration of hard bosons. Conversely, any configuration of hard bosons defines a unique loop configuration according to the following rules: (i) An isolated empty site corresponds to a double dimer; (ii) For two or more consecutive empty sites, there is one dimer on the external diagonal bonds and on all legs between them. These rules follow easily from the loop constraints and from the absence of dimer on the diagonal bond and on its associated leg bonds for an occupied site.

In terms of hard bosons, the first term of the QLM destroys or creates a boson on a rung and can be written as:

$$\left| \nearrow \nearrow \right\rangle \left\langle \overset{\bullet\!-\!\bullet}{\bullet\!-\!\bullet} \right| + \text{h.c.} = d_j^\dagger + d_j. \tag{14}$$

The second term is only non zero if the central rung bond has two dimers and the neighboring bonds have no dimer, i.e. if the central bond has no boson and the neighboring ones are occupied by one. It can thus be written in terms of hard bosons as:

$$\left| \mathbb{Z} \right\rangle \left\langle \mathbb{Z} \right| = n_{i-1}(1 - n_i)n_{i+1} = n_{i-1}n_{i+1}, \tag{15}$$

where the hard boson constraint has been used to write the last equality. Finally, if a rung is occupied by a single dimer, and since the Haldane configuration and its descendants are excluded from the Hilbert space, one of the neighboring rungs and associated legs will have to carry two bonds, and the other to be empty, so that the third term of the Hamiltonian can be rewritten as:

$$|\mathcal{J}\rangle\langle\mathcal{J}| \;=\; (1-n_{i-1})(1-n_i)n_{i+1} + n_{i-1}(1-n_i)(1-n_{i+1}) \;=\; 2n_i - 2n_{i-1}n_{i+1}, \quad (16)$$

where Eqs. 4 and 6 have been used in the last equality. Therefore, the quantum loop Hamiltonian of Eq. 13 can be written in terms of hard bosons as:

$$H_{\text{QLM}}^{\text{HB}} = \sum_j \left[ -J(d_j^\dagger + d_j) - 2\theta n_j + (2\theta - \delta)n_{j-1}n_{j+1} \right]. \quad (17)$$

Using the mapping of the QDM to the hard boson model, the quantum loop Hamiltonian of Eq.13 can be rewritten as a QDM as:

$$H_{\text{QLM}}^{\text{QDM}} = \sum_{\text{Plaquettes}} \left[ -J\left( |\updownarrow\updownarrow\rangle\langle\leftrightarrow| + \text{h.c.} \right) + (2\theta - \delta)|\updownarrow\updownarrow\rangle\langle\updownarrow\updownarrow| + (4\theta - 3\delta)|\leftrightarrow\rangle\langle\leftrightarrow| \right]. \quad (18)$$

So the QLM is equivalent to the QDM Hamiltonian of Eq. 1 with coupling constants $v_{\text{rung}} = 2\theta - \delta$ and $v_{\text{leg}} = 4\theta - 3\delta$.

## 2.4 Fibonacci anyon chain

Quite remarkably, the dimension of the constrained Hilbert space of the QDM and of the hard-boson model is equal to the Fibonacci number (see below), which is also the dimension of the Hilbert space of the Fibonacci anyon chain introduced by Feiguin et al [11]. It is thus natural to ask whether the two models are related. The Hamiltonian of the Fibonacci anyon chain can be defined in terms of the $\tau$-anyon occupation numbers $\tilde{n}_i$:

$$
\begin{aligned}
H_{\text{Fibonacci}} \;=\; & \sum_i \Big[ -\tilde{n}_{i-1}\sigma_i^x\tilde{n}_{i+1} + \varphi^{3/2}(\tilde{n}_{i-1} + \tilde{n}_{i+1} - 1) \\
& - \varphi^{-3/2}\tilde{n}_{i-1}\tilde{n}_i\tilde{n}_{i+1} - (\varphi^{3/2} + \varphi^{-1/2})\tilde{n}_{i-1}\tilde{n}_{i+1} \Big],
\end{aligned} \quad (19)
$$

where $\sigma_i^x$ is a Pauli matrix and $\varphi = \frac{1+\sqrt{5}}{2}$ is the golden ratio. The fusion rules lead to the constraint that at least one of two consecutive lattice sites is occupied by an anyon. It is thus very similar to the hard-boson constraint if one performs a particle-hole transformation and associates a hard boson with an empty site of the chain of anyons, and a $\tau$-anyon with an empty site of the hard-boson chain (see also Fig.2):

$$\tilde{n}_i = 1 - n_i. \quad (20)$$

The first term is a hopping term, and the factor $\tilde{n}_{i-1}\tilde{n}_{i+1}$ requires sites $i-1$ and $i+1$ in the hard-boson chain to be empty, so that site $i$ can be either occupied or empty and $\sigma_i^x$ just changes the occupation at site $i$. If, by contrast $\sigma_i^x$ is applied to a site $i$ such that at least one of the sites $i-1$ or $i+1$ is occupied, the result is equal to zero due to the hard-boson constraint. So in the hard-boson language this term is simply given by:

$$\tilde{n}_{i-1}\sigma_i^x\tilde{n}_{i+1} = d_i^\dagger + d_i. \quad (21)$$

The other terms are trivially translated according to the particle-hole transformation. The resulting Hamiltonian can be further simplified using the hard-boson constraint, leading to:

$$H_{\text{Fibonacci}} = \sum_j \left[ -(d_j^\dagger + d_j) + (3\varphi^{-3/2} + 2\varphi^{-1/2})n_j - (\varphi^{3/2} + \varphi^{-3/2} + \varphi^{-1/2})n_{j-1}n_{j+1} \right]. \quad (22)$$

Using the fundamental property of the golden ratio $\varphi = 1 + \varphi^{-1}$, this can be rewritten:

$$H_{\text{Fibonacci}} = \sum_j \left[ -(d_j^\dagger + d_j) + (\varphi^{5/2} - \varphi^{-5/2})n_j - \varphi^{5/2}n_{j-1}n_{j+1} \right]. \quad (23)$$

This is exactly the Hamiltonian of the tricritical Ising point of the hard-boson by Fendley et al. [10], while the Hamiltonian $\tilde{H}_{\text{Fibonacci}}$ obained by changing the sign of the potential terms is the Hamiltonian of the 3-state Potts critical point of that model (see below).

For completeness, we note that in terms of quantum dimer operators, the Fibonacci Hamiltonian takes the following form:

$$H_{\text{Fibonacci}} = \sum_{\text{Plaquettes}} \left[ -\left(|{\updownarrow}\,{\updownarrow}\rangle\langle\!{\cdots}\!| + \text{h.c.}\right) - \varphi^{5/2}|{\updownarrow}\,{\updownarrow}\rangle\langle{\updownarrow}\,{\updownarrow}| - (2\varphi^{5/2} + 3\varphi^{-5/2})|{\cdots}\rangle\langle\!{\cdots}\!| \right]. \quad (24)$$

## 3 Method: DMRG with quantum dimer constraint

Most of the numerical results in the present paper have been obtained with a Density Matrix Renormalization Group (DMRG) algorithm [15–17], while most of the earlier numerical investigations of quantum dimer models in the context of 2D models have been performed either with exact diagonalizations (ED) or Quantum Monte Carlo (QMC). When DMRG has been used, the quantum dimer constraint has not been encoded explicitly, but through an additional term in the Hamiltonian that penalizes energetically the states that do not satisfy the local constraints of the model [11, 18, 19]. In this section we explain how to explicitly implement the local constraint in a variational Matrix Product States (MPS) algorithm.

First, we define dimer creation and annihilation operators on bond $j$:

$$S_j^+ = \begin{pmatrix} 0 & 1 \\ 0 & 0 \end{pmatrix}, \quad S_j^- = \begin{pmatrix} 0 & 0 \\ 1 & 0 \end{pmatrix}. \quad (25)$$

A two-leg ladder with $N$ rungs is described by $3N-2$ variables on the bonds of the ladder. Each bond variable can be in one of two states: occupied or unoccupied. For convenience, the bond variables are labeled in the way shown in Fig.4.

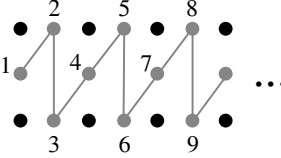

Figure 4: Labeling of the bond variables of the two-leg ladder.

The first term in the QDM Hamiltonian $|{\updownarrow}\,{\updownarrow}\rangle\langle\!{\cdots}\!|$ is equal to $S_j^+ S_{j+1}^- S_{j+2}^- S_{j+3}^+$ in terms of bond operators, where $j$ is the bond index of the left rung of the plaquette. The product operator $n_j = S_j^+ S_j^-$ counts the number of dimers on a selected bond $j$. For example, the last term of the Hamiltonian of Eq.1 is given by $|{\updownarrow}\,{\updownarrow}\rangle\langle{\updownarrow}\,{\updownarrow}| = n_j n_{j+3}$.

Matrix Product Operators (MPO) for the QDM Hamiltonians can be constructed using the standard procedure explained for example in Ref. [16]. The explicit form of the MPO's can be found in App.B.

As mentioned above, there are two constraints that distinguish quantum dimer models from conventional spin models: i) all sites on the original lattice belong to a dimer, so for a two-leg ladder the total number of dimers is exactly equal to $N_r$; ii) one site cannot belong to more than one dimer, so there are no corner-sharing dimers. The implementation of these two constraints in DMRG is rather advanced and to the best of our knowledge has not been explained in the literature. Below we provide a detailed explanation for the quantum dimer model. It is straightforward to generalize this approach to other types of local constraints. An example can be found in Appendix D, where an explicit algorithm is described for the QLM.

In ED the above constraints are imposed by removing from the basis of the full Hilbert space the vectors that do not contain the right number of dimers or that contain corner-sharing dimers. Then the Hamiltonian is diagonalized within this new basis using one of the standard routines, e.g. Lanczos algorithm. In DMRG the whole system is split into left and right environments and a central part, usually containing one or two sites. When all three parts are contracted and form a network usually called an effective Hamiltonian, the states that do not satisfy the QDM constraints must be removed and the effective Hamiltonian diagonalized in the reduced basis. Since the quantum dimer constraints are purely local, we propose to select the good basis states at each level of the algorithm: for the right and left environments we keep only states allowed by the QDM constraints, and the dimension of the physical bonds in multi-site MPO is also reduced by filtering out the states that do not satisfy the QDM constraints. Finally, the good basis states are further selected when contracting the effective Hamiltonian.

In order to implement the quantum dimer constraints in the left and right environments, we introduce the binary labels '0' and '1'. These labels distinguish two sectors of the Hilbert space, and thus can be treated as auxiliary quantum numbers of the model. For concreteness let us consider the construction of the left environment sketched in Fig.5. The subsystem with only one rung is labeled by '0' if it contains a dimer and by '1' if the bond is empty. If the first rung is occupied by a dimer, legs 2 and 3 remain unoccupied to satisfy the QDM constraints. By contrast, if the fist rung is empty, legs 2 and 3 are occupied by two dimers, otherwise the original site will remain unpaired. If the state of the left environment that contains the rung 1 and the legs 2 and 3 is labeled by '1', rung 4 cannot be occupied by a dimer, and the state becomes '0', whereas if it is labeled by '0' rung '4' can be either occupied or empty which leads to the states '0' or '1' respectively. Fig.5(b) summarizes the general rules for the construction of the basis. The generalization to the right environment is straightforward.

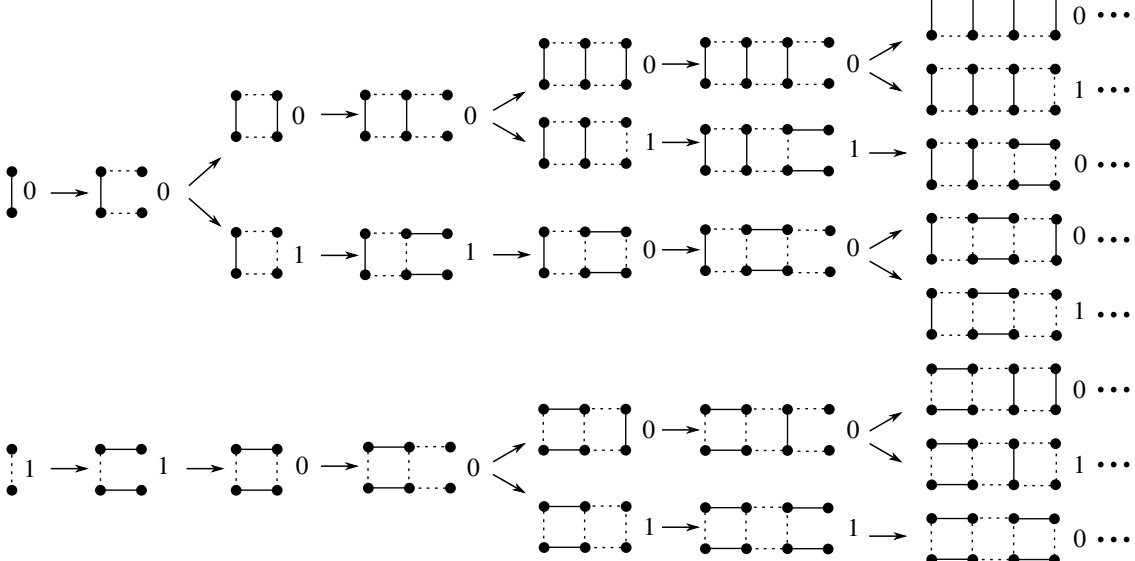

Figure 5: Construction of all possible states of the left block (see main text for details)

To summarize, the states of the left environment are labeled by '0' and '1' with the following rule: For a left subsystem that ends with a rung $j$, the left environment is labeled by '1' if there is no dimer either on the rung 'j' or on the legs $(j-1)$ and $(j-2)$. Otherwise, the left environment is labeled by '0'. States that terminate with legs are labeled by the same number as the last rung. This is true because on a ladder attaching two legs does not change the size of the Hilbert space of each sector. This implies that the MPS on the legs can be represented as a block-diagonal matrix that consists of only identity blocks and zero blocks.

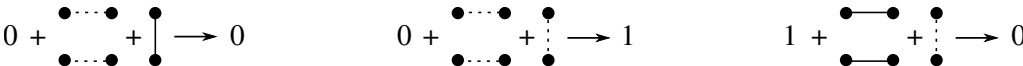

Figure 6: Fusion rules to construct left and right environments.

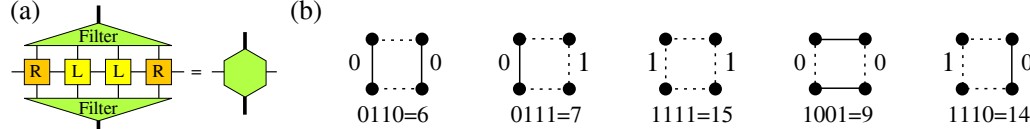

Figure 7: (a) Graphical representation of four-site MPO. The filter tensor allows to pass through only those states that are not forbidden by quantum dimer model listed in (b). The index of each configuration of (b) in the full basis of four-site Hamiltonian can be obtained using, for example, binary representation with "0" corresponds to a dimer and "1" to an empty bond as shown below each sketch

Importantly, the size of each sector '0' and '1' of the environments grows with the number of rungs $n$ as the Fibonacci series: $\Omega_0(n) = \mathcal{F}(n)$ and $\Omega_1(n) = \mathcal{F}(n-1)$, with $\mathcal{F}(0) = \mathcal{F}(1) = 1$. The total size of the Hilbert space for the left and right environments also grows as a Fibonacci series and is given by $\mathcal{F}(n) + \mathcal{F}(n-1) = \mathcal{F}(n+1)$. It is easy to check that in the system with $n_l$ rungs in the left and $n_r$ rungs in the right environments and with two-rung MPO shown in Fig.7(b) the total Hilbert space is given by:

$$\begin{aligned}
\mathcal{H} &= 2\mathcal{F}(n_l)\mathcal{F}(n_r) + \mathcal{F}(n_l)\mathcal{F}(n_r-1) + \mathcal{F}(n_l-1)\mathcal{F}(n_r) + \mathcal{F}(n_l-1)\mathcal{F}(n_r-1) \\
&= \mathcal{F}(n_l+n_r) + \mathcal{F}(n_l)\mathcal{F}(n_r+1) + \mathcal{F}(n_l-1)\mathcal{F}(n_r) \\
&= \mathcal{F}(n_l+n_r) + \mathcal{F}(n_l+n_r+1) = \mathcal{F}(n_l+n_r+2),
\end{aligned} \tag{26}$$

where we used the property of the Fibonacci numbers $\mathcal{F}(m)\mathcal{F}(n) + \mathcal{F}(m-1)\mathcal{F}(n-1) = \mathcal{F}(m+n)$, valid for series defined by the initial values $\mathcal{F}(0) = 1$ and $\mathcal{F}(1) = 1$. The explicit implementation of the QDM constraint thus allows us to reduce the size of the Hilbert space from $2^{3N_r+1}$ to Fibonacci number $\mathcal{F}(N_r)$, in agreement with Ref. [20]. This implies that the Hilbert space only grows approximately as $\Omega_0(N_r) \approx \varphi^{N_r} \approx 1.6^{N_r}$ instead of $\Omega(N_r) \approx 8^{N_r}$, the size of the Hilbert space of the model when the QDM constraints are replaced by terms that penalize energetically the states that do not fulfill the constraints.

It is worth mentioning that the approach we suggest is not the first attempt to target the reduced Hilbert space of a constrained model with tensor network algorithms. In the context of anyons, the anyonic fusion tree has been encoded explicitly first into MERA [21, 22] and more recently into TEBD [23, 24] through an auxiliary 'fusion tensor'. The role of this tensor is to select the states from the MPS tensor that satisfy the constraint and to discard all the other ones. Since in our approach we do not write the full tensor of the unconstrained Hilbert space but directly select blocks of allowed states (and perform the SVD decomposition on each block separately), the computational cost of the method presented here is lower. Due to braiding in Fibonacci anyons, there is a selected direction that suggests another way to implement an explicit local constraint. It has been proposed to associate a variable with the link shared by left and right environments so that each of them has a block-diagonal form with respect to this variable [25]. In quantum dimer ladders, we label the environment blocks by the states of the last rung which, according to our mapping, is located between or connects two Fibonacci anyons. In this sense our approach is similar to the one employed in Ref. [25]. However, we enforce the QDM constraint not only when diagonalizing the effective Hamiltonian but also at each local MPS tensor. This reduces the space necessary to store the wave-function (intermediate or final) and facilitates the following computation of observables.

Note that there is a fundamental difference between the implementation of the QDM and QLM (see App. D) constraint and the implementation of the hard-boson constraint into MPS. Since hard-boson degrees of freedom are associated with QDM plaquettes, or legs, then labeling the left or right environment by the state of its last site, '1' for site occupied with a boson and '0' for empty site, one can easily convince oneself that, while a block of states '1' of the left environment can only be connected to a block of states '0' of the right environment, a left block of states '0' can be connected to any state on the right. In this case, the tensors do not have a simple block diagonal form. Inspired by the implementation for the Fibonacci anyons [25], one can implement the hard-boson constraint by breaking the symmetry between left and right environments and use the states of one environment, e.g. the left one, to label the states of both environments. In this case the block-diagonal structure of each MPS tensor in the right environment cannot be preserved, but the hard-boson constraint can be satisfied explicitly when diagonalizing the effective Hamiltonian. To summarize, we find it much more convenient to use the quantum dimer version of the model to work out numerically the phase diagram of all models listed above, and this is what we have done.

In all simulations, we keep all states with Schmidt values larger than $10^{-12}$. We perform up to twelve sweeps and keep up to 2200 states. In order to calculate the excitation spectrum close to criticality we perform two or three additional sweeps without increasing significantly the number of kept states, during which several low-lying energies of the effective Hamiltonian are calculated. Further details of the method can be found in Ref. [26].

# 4  Quantum dimer model

## 4.1  Phase diagram in the hard-boson language

The ground-state phase diagram of the quantum dimer model defined by the Hamiltonian of Eq.10, i.e. using the natural variables of the hard-boson model, or equivalently of the hard-boson model of Eq. 2, is shown in Fig.8. It is drawn after Refs. [10] and [13]. It consists of three main gapped phases: a rung dimer phase, a period-three phase, and a columnar leg phase, and of a very narrow critical incommensurate phase along part of the boundary between the period-three phase and the disordered phases (see below). In the large-$U$ limit, the system is in the rung-dimer phase. It corresponds to the most flippable state and thus minimizes the energy of the second term. When the coupling constant of the next-nearest-neighbor plaquettes interaction $V + U/2$ is large and negative the system is in the columnar leg phase - every other plaquette contains two dimers on its legs as sketched in Fig.8. The translation symmetry is spontaneously broken and the ground state is two-fold degenerate. The phase transition between the columnar leg phase and the rung dimer phase is first order below the Ising tricritical point located at $U = -V + 1/V$ and $V = -\varphi^{5/2} = -\left[(\sqrt{5}+1)/2\right]^{5/2}$ and is continuous in the Ising universality class above it. The period-three phase corresponds to the least flippable state. It spontaneously breaks the translation symmetry, and in the thermodynamic limit (or on periodic clusters with a number of rungs multiple of 3), the ground state is three-fold degenerate.

The critical theory along the phase boundary between the period-three and the rung-dimer phase is probably the most subtle aspect of this phase diagram. There is an integrable line along which the transition has been shown to be in the 3-state Potts universality class [10]. This Potts point is located at $U = -V + 1/V$ and $V = \varphi^{5/2} = \left[(\sqrt{5}+1)/2\right]^{5/2}$. Away from this Potts point, there is a relevant chiral perturbation, and the transition has either to be in the chiral universality class proposed by Huse and Fisher [27], or to occur through an intermediate critical floating phase. Using Bethe ansatz, Fendley et al have proven that such a

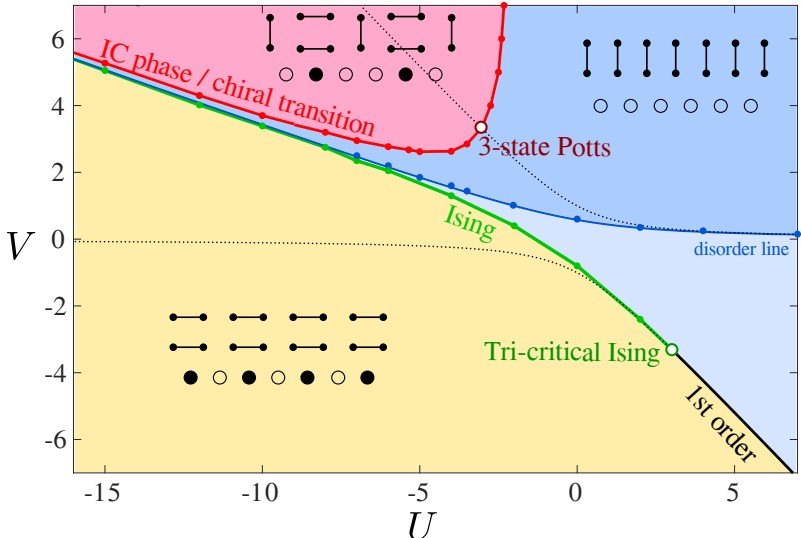

Figure 8: Combined phase diagram of the quantum dimer model defined by the Hamiltonian of Eq.10, and of the hard-boson model of Eq. 2 as a function of the natural coupling constants of the hard-boson model $U$ and $V$. It contains three gapped phases: the rung dimer phase, the columnar leg dimer phase, and the period-three phase. Sketches of the phases in both the QDM language (top) and hard-boson model (bottom) are included. The nature of the transition between the rung-dimer phase and the period-three phase changes along the critical line: (i) It is the three-state Potts universality class at the critical point (open dark red circle) on the integrable line (dotted line); (ii) In the vicinity of the integrable point the transition is chiral in the Huse-Fisher universality class [27]; (iii) Far from the Potts point the transition is through an intermediate floating phase in the Luttinger liquid universality class. The width of the critical phase is smaller than the width of the red line. The transition between the columnar leg and rung dimer phases is continuous in the Ising universality class (green line) above the tricritical Ising point (open green circle) and first order (thick black line) below it.

floating phase has to be present far from the Potts point in the limit $U \to -\infty$. More recently, Samajdar et al [12] have shown that, above the Potts point, the dynamical exponent is larger than 1, consistent with a chiral transition. Since then, the present authors [13] have used the algorithm of Section 3 to come with strong evidence that there is indeed an intermediate critical phase far from the Potts point, for large negative $U$ ($U < -4.5$), in agreement with Fendley et al [10], but also for large $V$ ($V > 6$). However, close to the Potts point, they found that the scaling of the wave-vector and of the correlation lengths are consistent with the chiral universality class of Huse and Fisher, in agreement with Samajdar et al [12] for the region above and not too far from the Potts point.

Finally, the disorder line that separates commensurate and incommensurate regions of the rung dimer phase has been determined numerically and is in excellent agreement with $U = -3V + 1/V$. The properties along this line have been studied analytically by Lesanovsky [28]. For $U \to -\infty$ the Ising critical line asymptotically approaches this disorder line.

## 4.2 Phase diagram in the QDM language

To make contact with the usual form of QDM easier, we show in Fig.9 the ground-state phase diagram of the Hamiltonian of Eq. 1 as a function of $v_{\text{rung}}$ and $v_{\text{leg}}$. As expected, in the limit $v_{\text{rung}} \to -\infty$ system is in the rung-dimer phase. For $v_{\text{leg}} \to -\infty$ the ground-state corresponds

to the columnar legs state. And the period-three phase is stabilized in the right upper corner of the phase diagram. There is an incommensurate critical phase between the period-three phase and the rung dimer for $v_{\mathrm{leg}} < 3.4$ and for $v_{\mathrm{leg}} > 19$, while between these values the transition is expected to be in the chiral universality class of Huse and Fisher. Therefore, in the isotropic quantum dimer model with $v_{\mathrm{rung}} = v_{\mathrm{leg}}$ the transition between the period-three phase and the rung dimer phase takes place through an intermediate critical phase with incommensurate correlations. The disorder line crosses the line of the isotropic QDM ladder at the Rokshar-Kivelson point. A detailed discussion of the phase diagram of the isotropic QDM can be found in App. C.

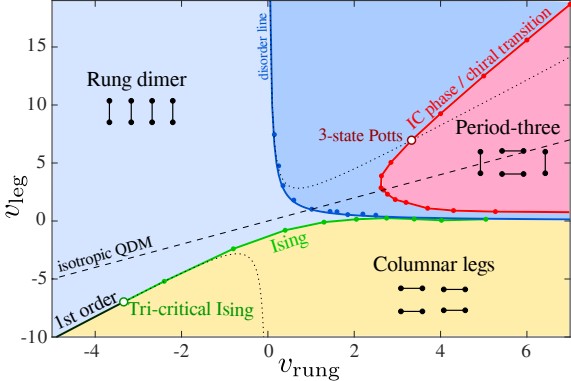

Figure 9: Phase diagram of the quantum dimer model Eq. 1 as a function of its natural coupling constants $v_{\mathrm{leg}}$ and $v_{\mathrm{rung}}$. It is just a reformulation of that of Fig. 8 with $v_{\mathrm{rung}} = V$ and $v_{\mathrm{leg}} = U + 3V$. The dashed line stands for the isotropic QDM defined by $v_{\mathrm{rung}} = v_{\mathrm{leg}}$ and studied in details in App. C.

## 4.3 Ising transition

### 4.3.1 Ising critical line

The transition between the columnar leg and rung dimer phases is first order below the tricritical Ising point and continuous above it. Fig. 10 provides numerical evidence in favor of Ising criticality at $V = 2$.

As an order parameter we take the dimerization on the legs $D(j, N_r) = \langle |n_{\mathrm{leg}}(j) - n_{\mathrm{leg}}(j+1)| \rangle$. In terms of hard-bosons this operator is defined by $D(j, N_r) = \langle |n_{\mathrm{HB}}(j) - n_{\mathrm{HB}}(j+1)| \rangle$. By looking at the finite-size scaling of the dimerization computed in the middle of the chain with open and free boundary conditions, we associate the critical line with the separatrix in the log-log plot as shown in Fig. 10(a). We also extract the critical exponent by looking at the profile of the dimerization in finite-size chains with fixed boundary conditions : hard bosons on the first and last site, or equivalently plaquettes with two leg dimers (see Fig. 10(b)).

The central charge has been extracted at the critical point from the scaling of the entanglement entropy with the block size in open systems. Following Ref. [29], we defined the reduced entanglement entropy $\tilde{S}_N(n)$ as the one with removed Friedel oscillations:

$$\tilde{S}_N(n) = S_N(n) - \zeta \langle \mathbf{S}_n \mathbf{S}_{n+1} \rangle, \tag{27}$$

where $\zeta$ is an adjustable parameter to whose value is chosen to best remove the oscillations. Then, according to CFT, the reduced entanglement entropy scales with the conformal distance $d(n) = \frac{2N}{\pi} \sin\left(\frac{\pi n}{N}\right)$ according to [30]:

$$\tilde{S}_N(n) = \frac{c}{6} \ln d(n) + s_1 + \log g. \tag{28}$$

The results agree within a few percent with the CFT predictions.

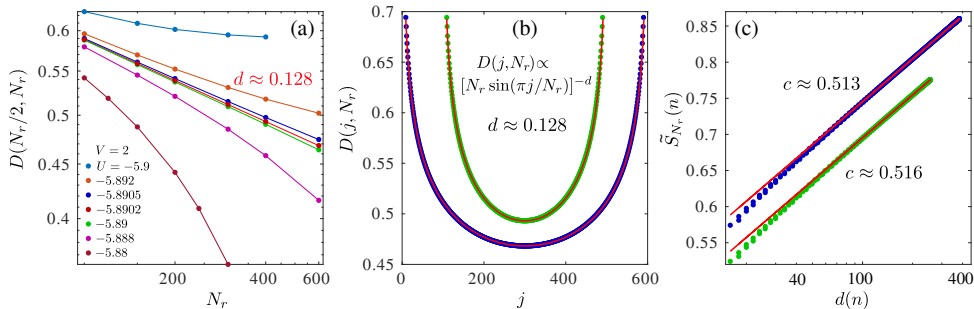

Figure 10: Numerical evidence in favor of an Ising transition in the hard-boson model. (a) Finite-size scaling of the dimerization computed in the middle of the chain with free boundary conditions. The quantum critical point for $V = 2$ corresponds to the straight line at $U \approx -5.8902$. The corresponding critical exponent $d \approx 0.128$ agrees within 3% with the Ising prediction $1/8$. (b) Decay of the Friedel oscillations away from the edges (see main text for boundary conditions). The critical exponent again agrees within 3% with the Ising prediction $1/8$. (c) Scaling of the reduced entanglement entropy with the conformal distance. The value obtained for the central charge agrees with the Ising prediction $c = 1/2$ within 4%

### 4.3.2 Boundary-field correspondence at the tricritical Ising point

The location of the tricritical Ising point is known exactly [10]:

$$V = -\varphi^{5/2}; \qquad U = -V + \frac{1}{V}. \tag{29}$$

The CFT prediction for the central charge is $c = 7/10$, and fixed boundary conditions to the state $\rightleftarrows$ at the first and last plaquettes are expected to induce Friedel oscillations in the number of dimers on the legs of the form $D(j) \propto \frac{1}{[N_r \sin(\pi j/N_r)]^d}$, where $D(j) = |\langle n_l(j) - n_l(j+1)\rangle|$, and $d$ is the critical exponent for the $\sigma$ spin operator given by $d = h_\sigma + \bar{h}_\sigma = \frac{3}{80} + \frac{3}{80} = 0.075$ for the Ising tricritical CFT. Our numerical results presented in Fig.11(a-b) are in excellent agreement with these predictions. In this section we take advantage of the exact location of the tricritical point and of this good agreement for simple boundary conditions to study the effect of the boundary conditions on the excitation spectrum of the tricritical Ising model, and to confirm numerically the CFT prediction for the boundary-field correspondence.

The correspondence between the primary fields and the boundary conditions for the tricritical Ising model has been worked out by Affleck [31]. It has been shown that fully polarized boundary conditions are associated with the identity and energy density conformal towers, for example $|\uparrow\rangle \longleftrightarrow I$ and $|\downarrow\rangle \longleftrightarrow \varepsilon''$ (the choice between the two is arbitrary), while unpolarized boundary conditions correspond to one of the spin conformal towers $|0\rangle \longleftrightarrow \sigma'$. By contrast to the Ising critical point, partially polarized boundary conditions are different from the fully polarized ones in the tricritical Ising CFT: $|0 \uparrow\rangle = \longleftrightarrow \varepsilon$ and $|0 \downarrow\rangle \longleftrightarrow \varepsilon'$. The correspondence between these notations and the original ones in Ref. [31] is the following: $|\uparrow\rangle = |(+)\rangle$, $|\downarrow\rangle = |(-)\rangle$, $|0 \uparrow\rangle = |(0+)\rangle$ and $|0 \downarrow\rangle = |(0-)\rangle$.

The correspondence between the boundary conditions of the hard boson model or of the quantum dimer model and those of the tricritical Ising theory is based on the mapping shown in Fig.2. We associate a quantum dimer ladder with dimers on the first pair of legs with $|\uparrow\rangle$ polarized boundary conditions. For the hard boson chain, this corresponds to occupied edge sites. Due to the quantum dimer constraint only every other plaquette can have a pair of leg

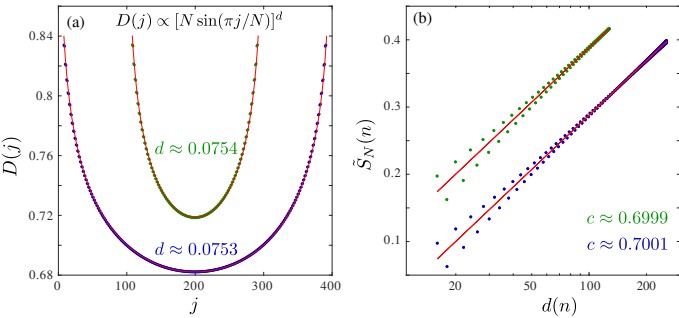

Figure 11: (a) Friedel oscillations at the tricritical Ising point induced by fixed boundary conditions with first and last sites occupied by bosons. (b) Scaling of the reduced entanglement entropy with the conformal distance $d(n)$. Green and blue dots are DMRG data for ladders with $N_r = 200$ and $N_r = 400$ rungs respectively. Red lines are the results of the fit with CFT predictions. In (b) the results for $N_r = 200$ are shifted vertically by $-0.05$ for clarity.

dimers. Equivalently, only every other site can be occupied by a hard boson. This implies that the quantum dimer and hard boson models at the tricritical point are equivalent to the antiferromagnetic Ising model. Thus, we associate the quantum dimer ladder with boundary conditions and even number of rungs with the $|\uparrow\rangle,|\uparrow\rangle \leftrightarrow I$ tricritical Ising tower and the ladder with boundary conditions and odd number of rungs with the $|\uparrow\rangle,|\downarrow\rangle \leftrightarrow \varepsilon''$ tower. According to Fig.12, these boundary conditions correspond to boson-boson in hard boson model, and to double dimer - double dimer in the quantum loop model. The boundary condition with a rung dimer on the fist rung is associated with unpolarized boundary conditions and the resulting towers do not depend on the number of rungs. It implies that the boundary condition corresponds to the $\sigma'$ conformal tower and that the boundary condition corresponds to a superposition of two towers $I + \varepsilon''$.

| Quantum dimer | Hard bosons | Quantum loop | Ising | CFT |
|---|---|---|---|---|
| ⊐,⊐ $N_r$ even | 1 , 1 $N_r$ even | //, // $N_r$ odd | ↑ , ↑ | $I$ |
| ⊐,⊐ $N_r$ odd | 1 , 1 $N_r$ odd | //, // $N_r$ even | ↑ , ↓ | $\varepsilon''$ |
| ⌶,⌶ | 0 , 0 | /, / | 0 , 0 | $I + \varepsilon''$ |
| ⊐,⌶ | 1 , 0 | //, / | ↑ , 0 | $\sigma'$ |
| ⊐(w),⊐(w) $N_r$ even | w1 , w1 $N_r$ even | //(w), //(w) $N_r$ odd | 0↑ , 0↑ | $I + \varepsilon'$ |
| ⊐(w),⊐(w) $N_r$ odd | w1 , w1 $N_r$ odd | //(w), //(w) $N_r$ odd | 0↑ , 0↓ | $\varepsilon + \varepsilon''$ |
| ⊐(w),⌶ | w1 , 0 | //(w), / | 0↑ , F | $\sigma$ |
| ⊐(w),⊐ $N_r$ even | w1 , 1 $N_r$ even | //(w), // $N_r$ odd | 0↑ , ↑ | $\varepsilon$ |
| ⊐(w),⊐ $N_r$ odd | w1 , 1 $N_r$ odd | //(w), // $N_r$ even | 0↑ , ↓ | $\varepsilon'$ |

Figure 12: Table of boundary field correspondence for the quantum dimer, hard-boson, and quantum loop models at the tricritical Ising point. Partially polarized boundary conditions are marked with $0\uparrow$ and $0\downarrow$ for Ising variables and with the letter 'w' and light gray color for weak effective boundary field.

The DMRG results for the energy spectrum of the quantum dimer model for different types of fixed boundary conditions are presented in Fig.13. They are in good agreement with the

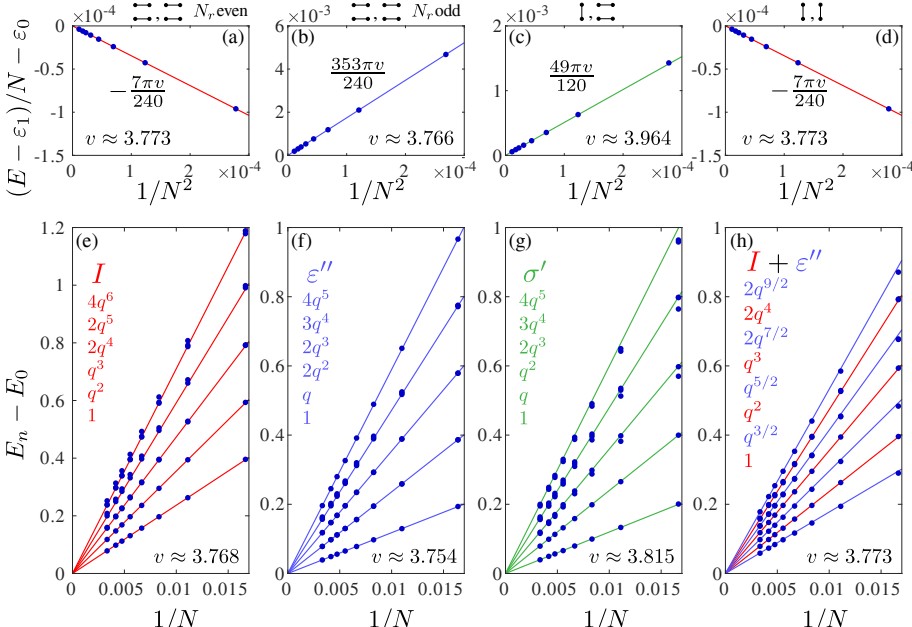

Figure 13: Finite-size scaling of the energy at the tricritical Ising point in the quantum dimer ladder with fixed boundary conditions. (a)-(d) Finite-size scaling of the universal term of the ground-state energy. (e)-(h) Finite-size scaling of the excitation energy with the inverse number of rungs. Blue dots are DMRG results. The velocities indicated in each panel are obtained by a numerical fit of the ground-state energy (a-d) or of the lowest excitation energies (e-h). Color lines are CFT predictions. The degeneracy of each level matches the CFT predictions, indicated as columns below the primary fields inside each panel.

CFT prediction.

Using fusion rules we have identified the conformal towers that appear in Ising chain with partially polarized or mixed boundary conditions:

$$|0\uparrow\rangle, |0\uparrow\rangle \leftrightarrow \varepsilon \otimes \varepsilon = I + \varepsilon'$$
$$|0\uparrow\rangle, |0\downarrow\rangle \leftrightarrow \varepsilon \otimes \varepsilon' = \varepsilon + \varepsilon''$$
$$|0\uparrow\rangle, |0\rangle = \varepsilon \otimes \sigma' = \sigma$$
$$|0\uparrow\rangle, |\uparrow\rangle = \varepsilon \otimes I = \varepsilon$$
$$|0\uparrow\rangle, |\downarrow\rangle = \varepsilon \otimes \varepsilon'' = \varepsilon'.$$

In order to determine the boundary conditions of the quantum dimer ladder that correspond to the partially polarized boundary conditions of the Ising chain, we have calculated the finite-size excitation spectrum as a function of the external boundary field $h$ applied on the first and last rungs. The excitation spectrum for $N_r = 240$ rungs as a function of the boundary field in the range from $-10J$ to $6J$ is shown in Fig.14(a). For large and negative field the energy spectrum corresponds to that with rung dimers at the edges and thus to free boundary conditions for the tricritical model. For high field the excitation spectrum is given by the identity conformal tower and corresponds to the fully polarized and aligned boundary conditions of the tricritical model. Between the two regimes the structure of the excitation spectrum is very different. It approaches the CFT prediction for partially polarized boundary conditions at $h \approx -2J$.

In Fig.14(b-k) we present the finite-size scaling of the ground-state energy and of the excitation spectrum with external fields that correspond to partially polarized boundary conditions.

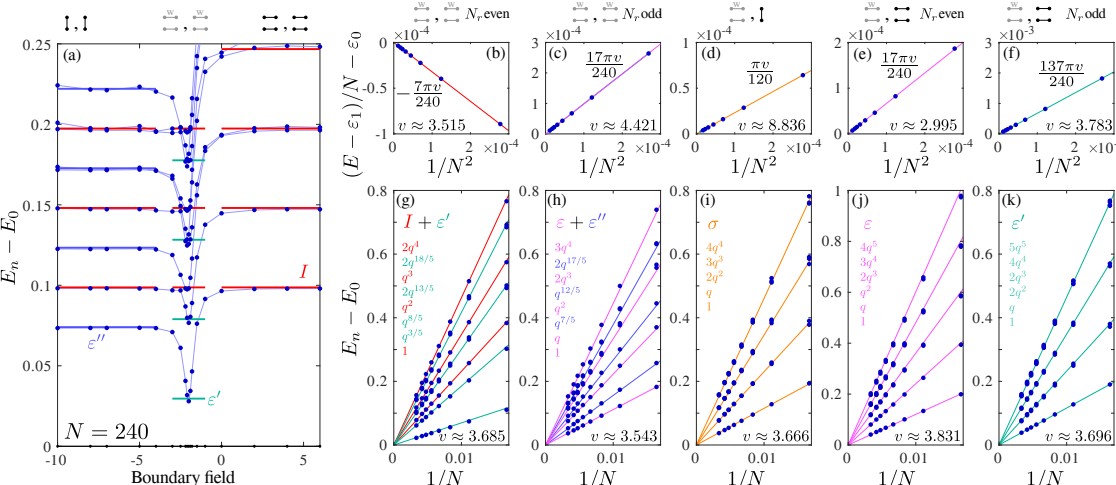

Figure 14: (a) Finite-size excitation spectrum as a function of the external boundary field applied at the first and last rungs of the ladder. Blue dots are DMRG data. Red, blue and green lines corresponds to the CFT prediction for I, $\varepsilon''$, and $\varepsilon'$ towers and $v = 3.77$. (b-f) Finite-size scaling of the ground-state energy for various boundary conditions, indicated on top of each panel. 'W' corresponds to the weak boundary field $h = -2$. (g-k) Finite-size scaling of the excitation energies with various boundary conditions. Blue dots are DMRG data. The velocities indicated in each panel are obtained by a numerical fit of the ground-state energy (a-f) or of the lowest excitation energies (g-k). Color lines are CFT prediction for partially polarized and mixed boundary conditions. The degeneracy of each level matches the CFT predictions, indicated as columns below the primary fields inside each panel.

The agreement between the CFT prediction for the conformal towers and the DMRG results for the excitation spectrum is excellent. By contrast the finite-size scaling of the ground-state energy is not as good. In particular, the velocity quoted in panel (d) seems to be off by a factor larger than 2, but this is probably due to the fact that the coefficient is very small, leading to a big uncertainty.

# 5 Quantum loop model

The ground-state phase diagram of the QLM of Eq.18 is presented in Fig.15. When both terms that favor dimerization and trimerization are small, the system is in the next-nearest neighbor (NNN) Haldane phase with a single dimer on every leg bond. When the coupling constant $\theta$ is large the system is in the trimerized phase. The transition between the trimerized and the NNN-Haldane phases is in the three-state Potts universality class through the integrable point located at $\delta = -\varphi^{-5/2}$ and $\theta = (\varphi^{5/2} + \varphi^{-5/2})/2$, it occurs through an intermediate floating phase for $\theta > 2.3$ and for $\delta < -4.7$, and it is a direct transition in the Huse-Fisher chiral universality class [27] otherwise. At large $\delta$, the system is in the dimerized phase. There is no direct transition between the dimerized and trimerized phase in this phase diagram. The transition between the NNN-Haldane phase and the dimerized phase is first order below the tricritical Ising point and continuous in the Ising universality class beyond it. In this phase diagram the tricritical Ising point is located at $\delta = \varphi^{-5/2}$ and $\theta = -(\varphi^{5/2} + \varphi^{-5/2})/2$.

By analogy with the QDM, the quantum loop constraint can also be encoded explicitly in MPS. All states can be grouped into three sectors labeled with different auxiliary quantum

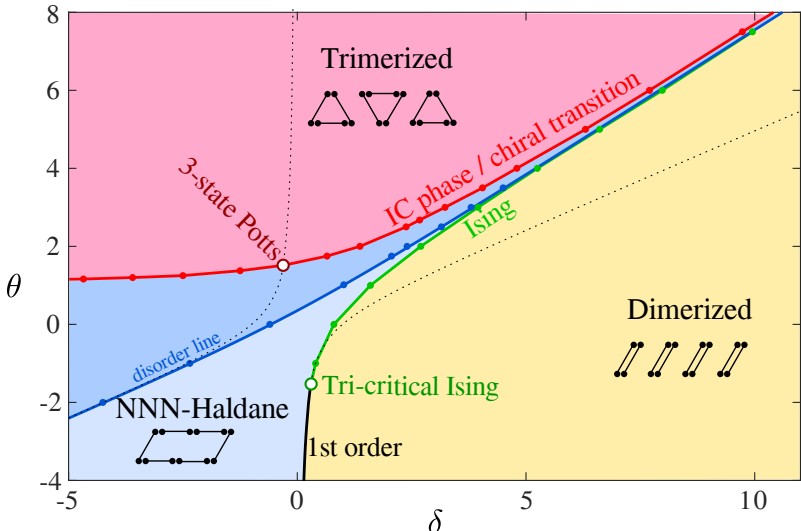

Figure 15: Phase diagram of the quantum loop model of Eq.13 as a function of $\delta$ and $\theta$. It has been deduced from the hard boson phase diagram shown of Fig.8 using $\delta = -U - V$ and $\theta = -U/2$

numbers. As in the QDM, there is a one-to-one correspondence between the quantum numbers of the left and right blocks, although in the quantum loop model the non-zero couplings correspond to the secondary diagonal. The details of the implementation are presented in Appendix D.

## 6  Conclusion

We have shown that the quantum dimer model on a two-leg ladder, the quantum loop model on a zig-zag ladder, and the hard-boson chain with next-nearest neighbor interactions can be rigorously mapped onto each other, defining a unique quantum model that lives in a constrained Hilbert space. This mapping has been instrumental in realizing that the quantum dimer and quantum loop models, for which very little was known, must have a rich phase diagram with period-two and period-three phases on top of a disordered phase, a result that has been established several years ago in the context of the hard-boson model [10].

Conversely, this mapping has led to a significant improvement in the numerical investigation of the hard boson model. Indeed, we have also shown that for the quantum dimer model (and in fact also for the quantum loop model) is it possible to implement simply and directly the quantum dimer constraint in a DMRG algorithm. This leads to a significant reduction of the dimension of the Hilbert space that allows one to perform simulations on systems with thousands of rungs. This algorithm has led to a better understanding of the nature of the transition between the period-three phase and the disordered phase of the hard-boson model [13].

Building further on the efficiency of this constrained DMRG algorithm, and taking advantage of the fact that the location of the Ising tricritical point is known exactly, we have also investigated numerically the correspondence between the boundary conditions of the quantum dimer ladder and the primary fields of the Ising tricritical point. In particular, we were able to drive the model into the regime that corresponds to the tricritical Ising model with *partially polarized* boundary conditions and to check numerically, to the best of our knowledge for the first time, the predictions of CFT for these boundary conditions.

We have also shown that the Fibonacci anyon chain can be rigorously mapped onto the

Ising tricritical point or the Potts critical point of these models depending on the overall sign of the potential terms of the Hamiltonian.

Coming back to the original motivation of this work, the investigation of effective models that describe transitions in spin chains and ladders that take place entirely in the singlet sector, we have shown that the QDM on a two leg ladder indeed has an Ising transition and can serve as a minimal model to study such transitions in spin-1/2 ladders, with the advantage that one can reach much larger systems than with standard DMRG with spin-1/2 degrees of freedom.

The generalization to a quantum loop model opens new perspectives. Indeed, beyond providing a very appealing picture of the Ising transition of the frustrated spin-1 chain between the NNN Haldane phase and the dimerized phase [6], it predicts that the transition between the NNN Haldane phase and a trimerized phase could be (i) chiral in the Huse-Fisher universality class; (ii) through an intermediate critical phase with incommensurate correlations separated from the gapped phase by Kosterlitz-Thouless and Pokrovsky-Talapov continuous transitions; (iii) direct in the three-state Potts universality class if the relevant chiral perturbation is absent; (iv) first order. Such a transition has been detected in the spin-1 bilinear-biquadratic zig-zag ladder [14], and it has been concluded that it must be first-order since the singlet-triplet gap does not close. It would be interesting to revisit this model to check for the presence of a gap closing in the singlet sector, and possibly of a chiral transition or of a critical incommensurate phase.

## Acknowledgements

We are indebted to Paul Fendley, Andreas Läuchli, and Steve White for insightful discussions, and to Guillaume Meyrat, who obtained some preliminary ED results in the context of his Master thesis. This work has been supported by the Swiss National Science Foundation. The calculations have been performed using the facilities of the Scientific IT and Application Support Center of EPFL.

## A  Mapping to Ising model

Although we have not used this mapping, it is worth mentioning that the quantum dimer model of Eq. 1 can also be mapped on a spin-1/2 Ising model in longitudinal and transverse fields according to the following rules (see Fig.2): all plaquettes that contain two dimers on the legs are assigned a spin $|\uparrow\rangle$, all other plaquettes a spin $|\downarrow\rangle$. In terms of these Ising variables, the Hamiltonian of Eq. 1 is equivalent to the $V_1 \to +\infty$ limit of the Hamiltonian

$$\tilde{H}_{\mathrm{QDM}}^{\mathrm{Ising}} = \sum_i \left[ v_{\mathrm{rung}} \left( S_{i-1}^z - \frac{1}{2} \right) \left( S_{i+1}^z - \frac{1}{2} \right) + (v_{\mathrm{leg}} - v_{\mathrm{rung}}) S_i^z - 2J S_i^x \right.$$
$$\left. + V_1 \left( S_i^z + \frac{1}{2} \right) \left( S_{i+1}^z + \frac{1}{2} \right) \right], \quad (30)$$

where the first term counts the number of flippable plaquettes, both with dimers on legs or dimers on rungs; the second term counts the number of plaquettes with two dimers on legs; the third term is off-diagonal and thus corresponds to the hopping term; finally, the last term has been introduced to satisfy the hard-core constraint in the $V_1 \to +\infty$ limit. Alternatively, the model can be simply defined by the Hamiltonian

$$H_{\mathrm{QDM}}^{\mathrm{Ising}} = \sum_i \left[ v_{\mathrm{rung}} \left( S_{i-1}^z - \frac{1}{2} \right) \left( S_{i+1}^z - \frac{1}{2} \right) + (v_{\mathrm{leg}} - v_{\mathrm{rung}}) S_i^z - 2J S_i^x \right] \quad (31)$$

acting in the constrained Hilbert space from which all configurations with neighboring ↑ spins are excluded.

## B   Matrix Product Operator

In this appendix, we provide a compact representation of the Matrix Product Operators (MPO) for the quantum dimer ladder defined by Eq.1. For clarity, zero entries in MPO are marked with dots.

$$
H_{\text{rung}} = \begin{pmatrix}
I & . & . & . & . \\
S^+ & . & . & . & . \\
S^- & . & . & . & . \\
S^+S^- & . & . & . & . \\
. & -JS^+ & -JS^- & v_{\text{rung}}S^+S^- & I
\end{pmatrix},
\tag{32}
$$

$$
H_{\text{leg}_1} = \begin{pmatrix}
I & . & . & . & . & . \\
. & S^- & . & . & . & . \\
. & . & S^+ & . & . & . \\
. & . & . & I & . & . \\
. & . & . & . & v_{\text{leg}}S^+S^- & I
\end{pmatrix},
\tag{33}
$$

$$
H_{\text{leg}_2} = \begin{pmatrix}
I & . & . & . & . \\
. & S^- & . & . & . \\
. & . & S^+ & . & . \\
. & . & . & I & . \\
S^+S^- & . & . & . & . \\
. & . & . & . & I
\end{pmatrix}.
\tag{34}
$$

## C   Quantum dimer line

To make contact with the literature on QDM on 2D lattices, where there is only one type of plaquette, we give a brief overview of the essential properties of the isotropic quantum dimer ladder defined by Eq.1 with $v_{\text{rung}} = v_{\text{leg}}$. The phase diagram is shown in Fig.16 and contains two gapped phases, a rung dimer phase, and a period-three phase, and a narrow critical incommensurate phase between them (the properties of each phase are described in the main text). The quantum phase transition between the rung dimer phase and the critical phase is expected to be in the Kosterlitz-Thouless universality class, while the phase transition between the critical phase and the period three phase is a continuous commensurate-incommensurate transition in Pokrovsky-Talapov universality class.

As for its 2D cousins, there is a Rokhsar-Kivelson (RK) point [1] at $J = v$ where the superposition of all dimer coverings with equal weight is a ground state. If we had not excluded the staggered states from the Hilbert space, a first order transition would take place at that point between the rung dimer phase and the staggered phase. With the exclusion of the staggered states, the model remains in the rung dimer phase up to $v/J \approx 2.67$. The RK point is nevertheless a special point: it coincides with the disorder point beyond which the short-range correlations become incommensurate.

For further reference in the general context of the QDM on various lattices, let us give some numerical results specific to this special case. The rung-rung correlations can be well fitted by

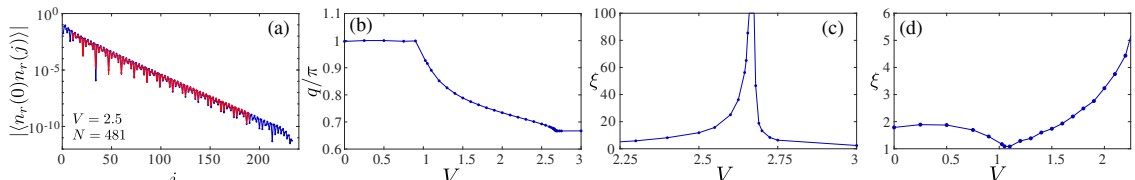

Figure 16: Phase diagram of quantum dimer model on two-leg ladder. Rung dimer phase is connected to the period-three phase via critical incommensurate phase. The width of the critical phase is smaller than the width of the red line. Short-range correlations in the rung dimer phase are commensurate before the Rokhsar-Kivelson point and incommensurate beyond.

the Ornstein-Zernicke form given by:

$$\langle n_{\text{rung}}(0) n_{\text{rung}}(j) \rangle \propto \frac{e^{-j/\xi}}{\sqrt{j}} \cos(qj + \varphi_0), \tag{35}$$

treating $\xi$, $q$ and $\varphi_0$ as a fitting parameters. An example of fit is shown in Fig.17(a). The resulting wave-vector is equal to $q = 2\pi/3$ in the period three-phase. Then, upon decreasing $v/J$, it slowly increases and reaches $q = \pi$ at the Rokhsar-Kivelson point $V = 1$ as shown in Fig.17(b). The coincidence of the disorder point with the Rokhsar-Kivelson point has been noticed earlier by Lesanovsky in the context of a model of strongly interacting one-dimensional Rydberg lattice gas model [32]. At this point, the correlation length has a kink - see Fig.17(d). This is a generic property of a disorder line [33, 34]. Around the transition at $V \approx 2.67$, the correlation length has a very pronounced peak (see Fig.17(c)).

Figure 17: (a) Example of the fitting of the rung-rung correlations to the Ornstein-Zernicke form of Eq.35 for ladder with $N_r = 481$ rungs, $V = 2.5$ and $U = -2V$. Blue dots are DMRG data, reg line are the result of the four-parameters fit. (b) Wave-vector $q$ as a function of $V$. (c-d) Correlation length as a function of $V$ has a pronounced peak around the critical point $V \approx 2.67$ (c) and a sharp minimum distinct for the disorder line at $V \approx 1$ (d).

The transition between the period-three and the rung-dimer phase takes place at $U \approx -5.358$ so inside the region where we believe the transition is through the floating phase. The numerical results are consistent with the presence of a very narrow critical incommensurate phase, but the finite-size effects are very strong because the wave-vector changes very little, and we cannot reach sizes where the Kosterlitz-Thouless and Pokrovsky-Talapov transitions would be clearly visible as separate transitions.

# D   DMRG with quantum loop constraint

The implementation of an MPS algorithm that takes into account the constraint directly for the QLM is also possible, and, as for the QDM but unlike the hard-boson case, it also leads to a simple formulation where left and right environments are in one to one correspondence.

To implement this algorithm, we follow the procedure described in Sec.3. We start with the construction of all possible dimer coverings of the left environments. By contrast to the single-dimer quantum dimer model, the two-dimer version requires three quantum numbers or labels. We label states with no unpaired dots by '0'. States that contain two unpaired dots are labeled by '1' if the two dots belong to two different sites and by '2' if the two dots belong to the same node.

The build-up procedure of the left environment is sketched in Fig.18. As in the case of the single-dimer QDM, the states of the legs are completely defined by the quantum label of the preceding rung: the leg is occupied by a dimer if the environment is labeled by '1', and remains empty otherwise. This implies that the on-site tensors on the legs consist of identity and zero blocks. Note that, quite importantly, the blocks of the left and right environments can be connected through the following rule: $(L,R) = \{(2,0),(1,1),(0,2)\}$. This implies that in order to connect two environments with a single square matrix of Schmidt values, the matrix has to be built out of three blocks placed on the secondary diagonal, and each block is a diagonal matrix with, as diagonal elements, the Schmidt values of the selected sector. The whole construction illustrated in Fig.18 can be summarized in the five fusion rules sketched in Fig.19.

Figure 18: (a) Construction of all possible states of the left block for the QLM (see main text for details)

$$0 + \overset{\cdots}{\diagup}{}^{\bullet\bullet} \longrightarrow 2 \qquad 1 + \overset{\phantom{\cdot}}{\diagup} \longrightarrow 0 \qquad 1 + \overset{\cdots}{\diagup} \longrightarrow 1 \qquad 2 + \overset{\phantom{\cdot}}{\diagup}{}^{\bullet} \longrightarrow 0 \qquad 2 + \overset{\cdots}{/\!\!/} \longrightarrow 1$$

Figure 19: Fusion rules for the quantum loop model on a zig-zag ladder with two dimers per node. A nearest-neighbor bond (rung) can be empty or occupied by one or two dimers, while a next-nearest-neighbor bond (leg) can be occupied by a single dimer or empty.

The matrix product operator of the QLM on a rung takes the following form:

$$H_{\text{rung}} = \begin{pmatrix} I_r & \cdot & \cdot & \cdot & \cdot & \cdot \\ \cdot & & \cdot & \cdot & I_r & \cdot & \cdot \\ \cdot & & \cdot & \cdot & \cdot & I_r & \cdot \\ r^\dagger & & \cdot & \cdot & \cdot & \cdot & \cdot \\ r & & \cdot & \cdot & \cdot & \cdot & \cdot \\ -\delta n_2 - \theta n_1 & -Jr^\dagger & -Jr & \cdot & \cdot & I_r \end{pmatrix}, \tag{36}$$

with

$$I_r = \begin{pmatrix} 1 & 0 & 0 \\ 0 & 1 & 0 \\ 0 & 0 & 1 \end{pmatrix}, \quad r = \begin{pmatrix} 0 & 0 & 0 \\ 1 & 0 & 0 \\ 0 & 1 & 0 \end{pmatrix}, \quad n_2 = \begin{pmatrix} 1 & 0 & 0 \\ 0 & 0 & 0 \\ 0 & 0 & 0 \end{pmatrix}, \quad n_1 = \begin{pmatrix} 0 & 0 & 0 \\ 0 & 1 & 0 \\ 0 & 0 & 0 \end{pmatrix}, \tag{37}$$

while the MPO on legs is diagonal:

$$H_{\text{leg}} = \begin{pmatrix} I_l & \cdot & \cdot & \cdot & \cdot & \cdot \\ \cdot & l & \cdot & \cdot & \cdot & \cdot \\ \cdot & \cdot & l^\dagger & \cdot & \cdot & \cdot \\ \cdot & \cdot & \cdot & l & \cdot & \cdot \\ \cdot & \cdot & \cdot & \cdot & l^\dagger & \cdot \\ \cdot & \cdot & \cdot & \cdot & \cdot & I_l \end{pmatrix}, \tag{38}$$

with

$$I_l = \begin{pmatrix} 1 & 0 \\ 0 & 1 \end{pmatrix}, \quad l = \begin{pmatrix} 0 & 0 \\ 1 & 0 \end{pmatrix}. \tag{39}$$

Note that the size of the local Hilbert space on a rung is of dimension three since it can be occupied by two dimers (trivial loop), a single dimer or no dimer at all, while the local Hilbert space on a leg bond is only of dimension two (empty or single dimer) since we have excluded states with double dimers on the leg bonds from the Hilbert space. As pointed out above, the leg tensors are trivial, so at each DMRG iterations we optimize two non-trivial rung tensors, and therefore run a three-site routine. As in the case of the single-dimer QDM, we contract the MPO of two rungs with a leg between them into a single large tensor and filter out the physical states that do not satisfy the QDM constraints. All possible states of the three-site Hamiltonian and the corresponding labels for the left and right environments are sketched in Fig.20

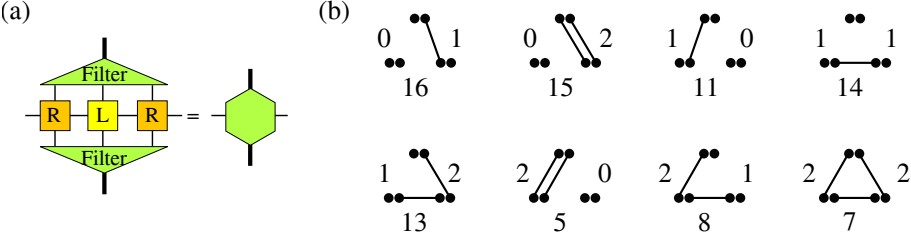

Figure 20: (a) Graphical representation of the three-site MPO. The filter tensor selects states that do not violate the QLM constraints. (b) Sketch of all possible states of the filtered three-site MPO in (a). The numbers on the left and right of each sketch indicate the quantum number of the left and right environments respectively. The number below is the index of each configuration in the full three-site Hilbert space (the first vector has index 0).

We have tested this algorithm on the phase diagram of the QLM, and as expected the results are identical to those deduced from that of the QDM and of the hard-boson model according to the mapping discussed in section 2.

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
