# Peer review of "DMRG investigation of constrained models: from quantum dimer and quantum loop ladders to hard-boson and Fibonacci anyon chains"

_SciPost Physics, doi:SciPost Phys. 6, 033 (2019)_

## Round 1 · Referee Report · Anonymous · 2019-1-18

Strengths

Clear, interesting

Weaknesses

Amount of new numerical results could have been larger

Report

The authors perform a DMRG study of constraint, one-dimensional models, in particular quantum dimer and quantum loop models on a ladder, as well as hard-boson and fibonacci anyon chains. Roughly, one can say that this paper has three different types of results. First, the authors establish a mapping between the four different models they study. Second, they show how one can directly implement the Hilbert space constraint in the DMRG algorithm. Direct implementation of the constraint gives of course a big advantage over using an enlarged (but more easily implementable) Hilbert space. Finally, the authors study the mentioned models via their DMRG algorithm.

The mapping between the various models is very clear, I do not have much to say about this, apart from the fact that these results are important, because they for instance shed light on the ladder quantum dimer model, because the hard boson model was studied in quite some detail before (refs [10,13] cited in the manuscript).

Section 3, where the method to implement the local Hilbert space constraint directly in the DMRG algorithm is also clear. The authors state that previous studies of constrained models by means of DMRG, implement the constraint by energetically penalizing the non-allowed configurations. However, it seems (little detail is given) that the DMRG study of the so-called `dilute fibonacci model' (section VII of PRB 87, 085106) uses a method that is similar to the one used in the present paper. If the authors agree, is seems appropriate to mention this. If not, this comment can be ignored.

Section 4 deals with the phase diagram of the quantum dimer model on the ladder. Using the known results in the literature (partially based on the authors' result [13] on the using the method described in sec. 3, on the nature of the phase transition between the period-three and rung-dimer phases), the phase diagram is discussed in detail. The new results presented in this paper are the analysis of the phase transition between rung-dimer and columnar-leg phases, where this transition is continuous and the analysis of the effect of the boundary conditions at the tri-crtitical Ising point. The authors show convincingly that the former is in the Ising universality class (though the finite size effects are apparently quite large, given the results for the central charge), and convincingly confirm the predictions by Affleck on the (partially polarized) boundary conditions at the tri-critical Ising points.

In conclusion, I think that this paper qualifies for publication in SciPost Physics. It would (indeed) have been interesting if the authors included results on the spin-1 bilinear-biquadratic zig-zag ladder, given that the amount of new, numerical results in the paper is not that large in comparison to the length of the paper. On the other hand, these results would most likely deserve a publication on their own.

Finally, I have some general remarks. The authors should carefully check the captions of the various figures, because they do not always match the actual figures. For instance, in fig. 1, there are four panels, but only three are mentioned in the caption, and they don't all match. In the last part of the caption of fig. 2, clearly something went wrong, and fig. 19 only consist of a caption, the actual figure is missing.

Requested changes

Figures and captions, see report

  • validity: top
  • significance: good
  • originality: high
  • clarity: high
  • formatting: good
  • grammar: good

Author:  Natalia Chepiga  on 2019-02-28  [id 452]

(in reply to Report 1 on 2019-01-18)

We thank the referee for his/her careful reading of the manuscript and for recommending this manuscript for publication in SciPost Physics. Let us briefly reply to the comments and suggestions in the referee’s report.

  1. It would (indeed) have been interesting if the authors included results on the spin-1 bilinear-biquadratic zig-zag ladder, given that the amount of new, numerical results in the paper is not that large in comparison to the length of the paper. On the other hand, these results would most likely deserve a publication on their own. This comment is well taken since indeed most of the numerical results that we used to confirm the chiral transition and the floating phase on the phase diagram have been published in a satellite paper (Ref.13). In the present manuscript, that we submitted to SciPost at roughly the same time as we posted Ref.13, our main goal was to provide details on the method and a discussion of several mappings. We agree with the Referee that it would be nice to present the results for the bilinear-biquadratic zig-zag ladder, and we are currently working on it. However, this project is numerically very heavy and will take about one year. We do not consider these results to be crucial enough to justify postponing the publication of the present manuscript for so long.

  2. However, it seems (little detail is given) that the DMRG study of the so-called `dilute fibonacci model' (section VII of PRB 87, 085106) uses a method that is similar to the one used in the present paper. If the authors agree, is seems appropriate to mention this. If not, this comment can be ignored. The DMRG used for the dilute Fibonacci anyon model is indeed similar yet a bit different from our approach. There the authors match the state from the left and right environments by including the middle bond simultaneously in both of them. This leads to an explicit anyonic constraint and a block-diagonal form of the effective Hamiltonian. In our approach, each MPS tensor satisfies the constraints, hence has a block-diagonal structure. However, in some sense the two methods are similar: we associate quantum numbers (labels) with rungs of the QDM ladder, and they correspond to a bond between two Fibonacci anyons. Note, that none of these two approaches can be directly applied to the hard boson model. We thank the Referee for this very relevant reference and added a brief discussion to the manuscript.

  3. The authors should carefully check the captions of the various figures, because they do not always match the actual figures. For instance, in fig. 1, there are four panels, but only three are mentioned in the caption, and they don't all match. In the last part of the caption of fig. 2, clearly something went wrong, and fig. 19 only consist of a caption, the actual figure is missing. We thank the Referee for pointing the mismatch in captions. We corrected the captions for Fig.1 and Fig. 2. However, we find that Fig.19 appears properly. It consists of the sketches of the fusion rules just above the caption.

---

## Editorial Decision

published